# A Protocol for a Mixed-Methods Process Evaluation of a Local Population Health Management System to Reduce Inequities in COVID-19 Vaccination Uptake

**DOI:** 10.3390/ijerph19084588

**Published:** 2022-04-11

**Authors:** Georgia Watson, Cassie Moore, Fiona Aspinal, Claudette Boa, Vusi Edeki, Andrew Hutchings, Rosalind Raine, Jessica Sheringham

**Affiliations:** 1London Boroughs of Camden & Islington, London N1 1XR, UK; Georgia.watson@islington.gov.uk (G.W.); cassie.moore@islington.gov.uk (C.M.); 2Department of Applied Health Research, University College London, London WC1E 7HB, UK; f.aspinal@ucl.ac.uk (F.A.); r.raine@ucl.ac.uk (R.R.); 3Public Health England, London SE1 8UG, UK; claudette.boa@phe.gov.uk (C.B.); vusi.edeki@phe.gov.uk (V.E.); 4London School of Hygiene and Tropical Medicine, London WC1E 7HT, UK; andrew.hutchings@lshtm.ac.uk

**Keywords:** population health management, data linkage, population health, inequalities, inequities, process evaluation, protocol

## Abstract

Population health management is an emerging technique to link and analyse patient data across several organisations in order to identify population needs and plan care. It is increasingly used in England and has become more important as health policy has sought to drive greater integration across health and care organisations. This protocol describes a mixed-methods process evaluation of an innovative population health management system in North Central London, England, serving a population of 1.5 million. It focuses on how staff have used a specific tool within North Central London’s population health management system designed to reduce inequities in COVID-19 vaccination. The COVID-19 vaccination Dashboard was first deployed from December 2020 and enables staff in North London to view variations in the uptake of COVID-19 vaccinations by population characteristics in near real-time. The evaluation will combine interviews with clinical and non-clinical staff with staff usage analytics, including the volume and frequency of staff Dashboard views, to describe the tool’s reach and identify possible mechanisms of impact. While seeking to provide timely insights to optimise the design of population health management tools in North Central London, it also seeks to provide longer term transferable learning on methods to evaluate population health management systems.

## 1. Introduction

In many countries, health policy has moved towards greater integration between different organisations that plan, commission, and deliver health and care [1]. In the latest stage of policy reforms in England, for example, all areas were statutorily required to form integrated care systems (ICSs) by April 2021 that include hospital, mental health, and community trust healthcare providers; primary care providers; clinical commissioning groups; and local authorities, which have a lead role for public health [2].

Sharing patient information across organisations is recognised as a key part of health system integration. An evaluation of four international integrated care systems conducted by the Nuffield Trust describes ‘informational integrative processes’ as one of the six key factors in the success of, or difficulties in, these programs. The challenges of data sharing across organisations, both in the United Kingdom and internationally, have been well documented [3]. However, a number of data sharing systems are now being developed and deployed. For example, the Whole Systems Integrated Care (WSIC) system in North West London, England was set up in part to facilitate the journey towards integrated care systems [4].

It is increasingly accepted that data sharing in itself is not sufficient to drive integration, and in turn improve population health outcomes [5]. As Scott et al. argue, ‘data alone does not save lives. It is knowledge derived from data analysis and applied in practice that saves lives’ [6]. Population health management (PHM) is an emerging technique used by local health and care partnerships in England. It uses data to help practitioners to understand their population, and then to use this understanding to inform practice [7]. It involves linking and analysing health and care data from different organisations to understand the health of a local population and their current service need, and to predict what local people will need in the future. In a PHM approach, this information is then used to inform decisions on the design and delivery of services in order to improve the health and wellbeing of the population and reduce inequities [8]. 

There is a lack of evidence about how (or whether) PHM can achieve its aims. There is some evidence that the use of data sharing platforms could influence patient care, but this is mainly from case studies, where evaluation has often been conducted in-house [9,10]. Before considering evaluating the effectiveness of such tools on population health, we need to know more about how PHM data sharing platforms may enable a population health management approach, i.e., to inform decisions to improve the health and wellbeing of their population and reduce inequities. This information is needed to optimise the design of PHM platforms and programs and inform impact evaluations in the future. 

Process evaluations—which seek to describe how, and why, a program or intervention works—are often used alongside impact evaluations [11]. For emerging interventions, they can also serve a useful purpose to understand key aspects of delivery of an intervention under development. This protocol describes a process evaluation of a population health management system in London, England using quantitative and qualitative methods. It has dual aims:‑To understand how staff use a specific population health management system tool to inform decisions and ways of working that reduce inequities in order to help the local area further develop their system;‑To develop the capacity for wider evaluation of PHM systems.

### 1.1. Context: The Population Health Management Innovation

This evaluation is centred on North Central London’s Integrated Care System (North London Partners, NLP), which provides care to 1.5 million people across five boroughs of London. North Central London is comparatively well advanced in its deployment of a near real-time population health management tool that integrates health and care data from across the system.

NLP uses HealtheIntent, a PHM platform developed by the digital provider, Cerner, which combines data across the 28 local authorities and health and care organisations in the integrated care system. It links and standardises data from across the health and social care system (such as general practices and hospitals) and re-presents these data—as ‘registries’ and ‘dashboards’—to staff based in the constituent organisations. An aim of NLP’s PHM program is to identify and reduce health inequities and, consequently, elements of HealtheIntent are specifically designed to assist users to identify segments of the population with unequal access to care to inform the development of targeted responses [7]. The design of the tools is underpinned by several core principles, including the relevance of intersectionality (i.e., recognising that social characteristics, such as ethnicity, gender, and socioeconomic circumstances, are interconnected and can create distinct, and sometimes amplified, experiences of disadvantage), the conceptualisation of health inequities as existing on a gradient rather than as a binary (present or absent), and the roots of inequities being in material and psychosocial factors upheld by political and economic structures. 

One of the first HealtheIntent tools used in NLP was a COVID-19 vaccination dashboard. Nationally, COVID-19 vaccinations were rolled out in phases. In the first phase, starting from 8 December 2020, the target was for all adults over 65 years of age, those in care homes, NHS and care staff, and clinically vulnerable people to have been offered a first vaccine dose by 15 February 2021 [12]. The second phase of the vaccination rollout, from 13 April 2021, covered the population aged 18–64 years and maintained priority by age and clinical risk [13]. As with many vaccination programs, there was concern that inequalities in uptake would result in inequalities in the risk of COVID-19 infections and serious sequelae. The Scientific Advisory Group for Emergencies (SAGE) reported that, in previous roll outs of national vaccine programs, there was lower uptake in minority ethnic populations [14]. Given the ethnic inequities in COVID-19 death rates, there were specific concerns about ethnic inequities in COVID-19 vaccine uptake [15].

The HealtheIntent COVID-19 vaccination Dashboard (referred to in the rest of the paper as the Dashboard) was developed at the end of 2020 (Figure 1). It sought to enable staff to view variations in COVID-19 vaccination uptake almost in real time. It became available to end users (NLP staff) in December 2020 and continues to be developed, updated, and improved in response to changing requirements. This has led to many iterations of the Dashboard, but, at the time of this evaluation, the Dashboard contained an overview page, a page describing uptake by eligibility cohorts, several demographic and equalities pages, data quality pages, a case-finding tool, and a user guide. 

Users have access to different versions of the Dashboard depending on their staff role and type. All users, including non-clinical staff, can see anonymised, aggregated data, but only those with permission, such as primary care staff, can access individual patient data. An Overview page describes overall vaccination uptake. An Equalities and Demographics page segments (i.e., enables users to stratify) the population by gender, ethnicity, IMD quintile, first language spoken, age, and geography. An example of segmentation is by the level of deprivation experienced. The Dashboard stratifies the population into five deprivation quintiles, in line with evidence about health inequalities existing on a gradient [16]. While this design cannot guarantee that users of the tool focus on the middle quintiles as well as the lowest quintile, it does provide users with the capability to do so and respond to findings.

The Dashboard tool also allows users to tailor what they view by providing filters (i.e., restricting the view to specific sub-populations). The Dashboard’s many filters include the user view (where users can limit what they see to their own care team type or organisation), COVID-19 information (vaccination eligible cohort, number of doses received, vaccine manufacturer), and health and care information (carer status, known to adult social care, long term conditions, number of long term conditions, mental health conditions, homelessness, bedbound and housebound status). The demographic variables displayed in charts on the Demographics and Equalities pages can also be used to filter data. The system is designed to prevent presenting data in numbers so small that identification might be (theoretically) possible to those without permission to access identifiable data.

### 1.2. Objectives

In order to address our first aim of understanding how a population health management system is used, we have proposed two objectives: To describe how (or whether) staff report using evidence of inequities in uptake available in the HealtheIntent COVID-19 Vaccination Dashboard to address inequities;To describe staff usage of the HealtheIntent COVID-19 Vaccination Dashboard, particularly those parts of the Dashboard that display evidence of inequities in uptake.

To address our second aim of building capacity for wider evaluation of PHM systems, there are two objectives to equip public health practitioners, working as embedded researchers in NLP, with the skills to undertake with supervision both the qualitative and the quantitative arms of the study. 

To reflect on the suitability of our methods, in particular we work closely with and train locally embedded researchers to determine the extent to which this model is a workable model for future evaluations of population health management.

## 2. Materials and Methods

This study will combine qualitative methods to identify potential mechanisms of impact of the Dashboard and quantitative analysis of Dashboard usage and reach. 

To support part of the study’s capacity building aim, university researchers will be working in collaboration with public health practitioners who are seconded part-time to a research role, funded by a grant intended to build capacity for public health research in local authorities. The seconded practitioners’ main public health roles are within public health teams in local and regional government. The practitioners will gain transferable skills in research and evaluation through access to specific courses and seminars (e.g., in evaluation methods) and through undertaking all stages of the research process, from submission for ethical review to dissemination of findings, with supervision and guidance from university researchers. If this objective is fulfilled, it will equip the seconded practitioners with the experience and skills to undertake evaluation of NLP’s PHM tools in the future. It will also advance our understanding of how such collaborations could be used to conduct evaluations of other PHM systems. 

The evaluation is designed to be relatively rapid (i.e., completed within 6–12 months) to ensure the findings are timely enough to influence future local population health management innovations. Therefore, we will incorporate the following elements of rapid evaluation approaches: multiple researchers collecting data concurrently; and sharing interim findings with stakeholders to shape interpretation and analysis, and to sustain their involvement and support [17]. 

The study was granted approval by UCL Ethics Committee, ref: 2037/005. We started activities for the evaluation in September 2021. We envisage completing most stages of the evaluation by the end of June 2022, though further analysis of the dataset may be undertaken after this date.

### 2.1. Proposed Qualitative Data Collection and Analysis

We will undertake semi-structured interviews (*n* ≈ 20) online using MS Teams with a purposive sample of staff who have responsibility for an aspect of COVID-19 vaccination planning or delivery.

Study population: We will interview staff at different levels of seniority, in clinical, strategic, commissioning, and analytical roles across different organisations in NLP (primary care, hospital or mental health providers, social care, public health), and will seek to ensure we capture experiences across all five North Central London boroughs. The sample size is approximate because some individuals will have more than one role, and thus will be able to cover more than one of our desired attributes. 

Interviews will explore staff experiences of using the Dashboard and how variations in vaccination uptake shown in the Dashboard informed their actions to address inequities. We have developed a topic guide (an example provided as Appendix A) informed by normalisation process theory, which provides guidance for exploring the perceptions of staff and the actions that staff take when a new product or innovation is introduced into an organization [18]. In line with normalisation process theory, the interviews will explore the following:‑Motivation to use the Dashboard;‑Specific features of the Dashboard and their advantages or limitations for the user;‑Contextual enablers or barriers to using the Dashboard;‑How the participants considered that usage of the Dashboard influenced ways of working and decisions, e.g., about COVID-19 vaccination planning or delivery.

To develop the guides and to develop consistency between interviewers, an exercise of ‘concept mapping’ was undertaken by the lead interviewer (G.W.) with supervision from F.A., whereby, for each topic covered by the guide, a short ‘concept’ description was developed to guide interviewers on what the question was seeking to obtain. This led to revisions of the interview schedule and enabled each interviewer to tailor their own guide to their own language and style. The guide was initially ‘soft’ pilot tested with a colleague and then, after refinements, was piloted with three staff working in North Central London. No significant changes were made to the guide at this point, so these interviews will be included in the final dataset.

Interviews will be conducted by several individuals (G.W., C.M., V.E., C.B.) working within and external to NLP, to expedite data collection. All interviewees will be asked to sign a consent form before being interviewed. Participants’ names and roles will not be disclosed and all data will be anonymised to minimise the risk of identifying participants. All interviews will be recorded and transcribed in full by a transcription service. The transcribers will remove any identifiers such as names and organizations before securely returning transcripts to the researchers. Researchers will read and further redact transcripts if any potentially identifiable information remains in the text. To expedite analysis, interviewers will note key points from their interviews immediately after conducting them. 

Transcripts will be analysed using the Framework Method using Excel by G.W. and C.M. with reading of selected transcripts by J.S. and F.A [19]. A preliminary coding framework drawn from the topic guide was developed by G.W. and C.M. in discussion with J.S. and F.A. to expedite initial descriptive analysis. Further codes and overarching themes and refinements to the analytical strategy will be generated inductively. Discussions with the wider team will take place to discuss emerging findings and resolve discrepancies in coding and interpretation of the data. 

Documents and correspondence about the Dashboard, including descriptions of the rollout of vaccination in NLP and iterations of the Dashboard, will be examined to provide contextual evidence for the interviews and to build a timeline of key events in the program to inform both qualitative and quantitative analysis (see Combining Qualitative and Quantitative Data). 

### 2.2. Proposed Quantitative Data Extraction and Analysis

The proposed quantitative data collation and analysis part of the study will use anonymised staff usage data already stored within HealtheIntent to describe variations in usage of the COVID-19 vaccination Dashboard since its launch in December 2020. It seeks to capitalise on the extensive data automatically generated about usage whenever these population health management tools are used. A request for anonymised data has been submitted to the HealtheIntent service desk. This request includes the numbers of staff by organization and over time that are registered to use the Dashboard. It also includes a request for figures on the actual use of the Dashboard, both in terms of logins and activities while on the Dashboard.

Initial descriptive analysis of usage will be undertaken in Stata and will involve two components [20]. First, the analysis will seek to enumerate the denominator population (i.e., the number of accounts of individuals that were registered to use the COVID-19 Dashboard) and its characteristics (e.g., organisation and geographical area). Second, the proportion of those using the Dashboard among those registered will be generated in key time periods (informed by the timeline constructed, see Proposed Qualitative Data Collection and Analysis section). Usage will be examined for any part of the Dashboard. Where possible, usage will be examined for specific equalities pages of the Dashboard and among specific groups of users, defined by organisation, staff role type, and geographical area.

### 2.3. Combining Qualitative and Quantitative Data: Proposed Approach

As described above, we have planned to use the qualitative and documentary data to construct a timeline of key events that will inform the intervals for the quantitative analysis. Interim findings from the qualitative and quantitative data will be shared within the study team at regular intervals, to inform the interpretation of findings from each method, and potentially to prompt further analysis. For example, interview data that reports barriers to, or motivations for, usage may be used to support interpretation of quantitative data showing variations in usage patterns. The extent to which it is possible to combine qualitative and quantitative findings will depend on the data obtained. We will seek to use both qualitative and quantitative findings to support the development of candidate program theories by which population health management could achieve its intended outcomes that could be used in future impact evaluations.

## 3. Discussion

This protocol describes a process evaluation of a specific population health management tool within one geographical area of England. It will combine qualitative and quantitative methods to describe staff usage of a specific tool, the COVID-19 vaccination uptake Dashboard, and how it informs their decisions and ways of working to reduce inequities in vaccine uptake. Working with colleagues based in North Central London means that any learning gained even in the earliest stages of the process evaluation can be rapidly fed back to inform continuing Dashboard development and new population health management tool development and rollouts. The findings from the study also have the potential to have wider significance in advancing methods for evaluating population health management, and thus could build capacity for further evaluations of population health management programs.

### Strengths and Limitations

A key strength of this evaluation is the collaboration between academia, local public health, health care, and regional public health teams. The use of embedded local researchers combined with senior sponsorship promises to ensure the evaluation remains grounded in local service priorities and serves to build local evaluation capacity. The timeliness of the evaluation, and sharing of preliminary findings, aligns with the principle of continuous learning and improvement underlying NLP’s PHM program and, more specifically, its use of linked data to support health and care providers addressing inequities. Regional public health input has brought a wider policy perspective and academic input brings independence and objectivity to the evaluation and provides methodological rigour.

The evaluation is subject to some important limitations or challenges. It is taking place in 2021 and 2022, at a time of considerable uncertainty owing to the COVID-19 pandemic. Therefore, it is possible this will affect access to interview participants and access to quantitative data. Strategic input from internal project sponsors will be sought to address barriers and encourage participation to reduce the risk of the project stalling due to other priorities. In the ongoing qualitative data collection and analysis, rapid evaluation approaches were chosen to enable timely findings and feedback to NLP and will also be subject to further in-depth analysis.

We anticipate two major challenges in the quantitative aspect of the evaluation. Access to data held within the population health management system by an external partner, such as a university-employed researcher, would require extensive information governance procedures, reducing the timeliness of the evaluation. However, all organisations within NLP contribute data to the system and are designated data controllers. This designation enables all partners access to non-identifiable data, which makes internal evaluation a possibility. To make use of the opportunity for internal analysis, a local analyst in a funded embedded researcher role will undertake the analysis with the support of external quantitative expertise from ARC North Thames. In addition, the data on staff usage have not previously been subject to evaluation or monitoring. It is thus not well understood what information is feasible to extract from the system, what processes are required to make this information suitable for analysis, or how best to do this. Therefore, the evaluation also seeks to clarify the range of data available, the processes for data extraction, and management before analysis. We will also iterate what data we request and develop a more detailed analysis plan as our understanding of the data evolves.

## 4. Conclusions

This protocol describes an evaluation that seeks to understand how staff use a specific population health management system tool to inform decisions and ways of working that reduce inequities in vaccine uptake. In the short term, achieving this aim should serve the local health and care system by providing useful insights to inform future population health management activities. The evaluation also aims to develop the capacity for wider evaluation of PHM systems. We will have met this aim if our evaluation equips local practitioners with the skills to conduct further evaluation and if it generates transferable learning about the methods for evaluating such programs in collaboration with local health and care professionals in the future.

## Figures and Tables

**Figure 1 ijerph-19-04588-f001:**
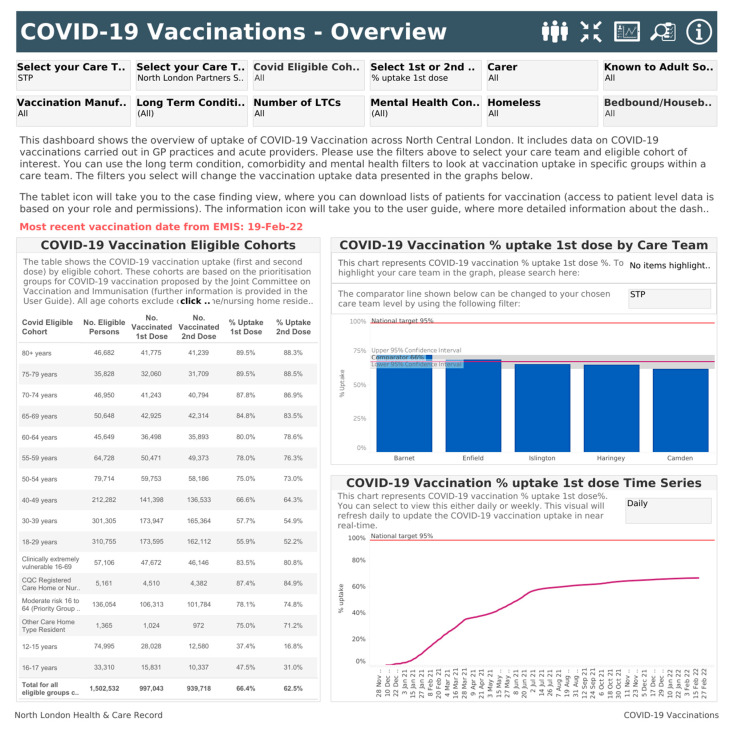
Illustrative screenshot of the HealtheIntent COVID-19 Vaccination Dashboard. Copyright: North London Partners.

## Data Availability

Not applicable.

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
