# Peer review of "A Protocol for a Mixed-Methods Process Evaluation of a Local Population Health Management System to Reduce Inequities in COVID-19 Vaccination Uptake"

_ijerph, 2022, doi:10.3390/ijerph19084588_

Round 1
Reviewer 1 Report
The authors provide great detail to the importance of the PHM as an integrated method for informing patient care and broader population needs. The primary objectives of this proposed evaluation are to describe the use of the NLP dashboard (PHM platform) as an evidence base for decision-making with respect to addressing inequities; to describe how the dashboard can be useful for identifying difference in patient uptake, access, and outcomes; and to develop methods that can inform praxis across NLP network.
This paper has many notable strengths as addressing this topic has critical implications for establishing and supporting an evidence base that can inform integrated care. Overall, the manuscript provides a conciseness of style in presentation and is well-written but can be improved upon with minor adjustments to sentence structure. With these strengths in mind, the following points are offered to assist in the overall quality of the written product.
- Broadly the authors should consider replacing their use of the term inequality with inequity. Inequalities relate to the uneven distribution of resources and it seems that some of the challenges within healthcare are more related to inequity with a focus on systemic barriers that exacerbate unfair, and avoidable differences that patients may be experiencing as it relates to social factors, education, employment, gender, ethnicity etc.
- This research is proposed and has not been conducted to date, so the language should always be future tense. The abstract reads as if the research is underway and it would be helpful for this to be clarified.
- line 66 - Please clarify the use of RCTs in this example. I think the authors are trying to use an analogy of process and impact evaluation being similar to RCTs with clinical research but the relationship is not clear.
- Line 96 - When describing the vaccine dashboard rollout, the term "from" implies that there will also be a "to" as a booked of time, it would be useful to revise this sentence.
- Lines 114-116 and 136-138 are duplicated text and should be revised.
- Line 141 - A transition statement would be helpful prior to the listing of objectives.
- The Materials/Methods section need a subsection heading for the second paragraph as it relates to the study population and proposed data collection and should be properly labeled.
- All Sections under Materials and Methods should be labeled as "proposed" analysis/es.
- Line 168 has a typo preceding reference 17, using "." instead of a ":".
- It is strongly suggested that the proposed qualitative methods/analysis section be enhanced. Qualitative methods should detail the following: provide reference to the pre-testing or pilot testing of the survey instrument and discussion guides; detail if interviews will be transcribed or summarized; detail the development of a codebook; how many persons will comprise the coding team with detail about interrater reliability and coding agreement; and reference the use of a data management analytic software (e.g. NVivo, MaxQDA, etc.).
- Section 2.2 reads "this component" and could benefit from a rephrase as a starting sentence that details the proposed quantitative data analysis portion of the methods.
- Section 2.3 could benefit from additional detail as to how the data will be combined. Please clarify if a mixed methods analytic software will be utilized to manage the data integration or if another form of mixed methods thematic/pattern analysis will inform how the data are merged.
Author Response
Thank you to the reviewer for extremely helpful comments. We have attached a point-by-point response here.

Reviewer 2 Report
I welcome a publication describing the publication of the collaboration between academia, local service public health, health care services and regional public health teams. The methods are appropriate and described in detail.
The protocol appears to indicate that the study is already in progress, and may even have been completed as of March 2022. If the study has not yet been carried out, some clarity around the phrase "It is taking place in 2021/22."
I can see how these findings will be useful to those working in the public health system in England. However, international applicability may be minimal. I recommend that authors seek a review from somebody less familiar with the English system, to ensure concepts discussed can be understood by those outside of the UK.
Author Response
Thank you to the review for this helpful review. We attach our point by point response here.

Reviewer 3 Report
Title:
Title is too long.
it’s advisable to avoid questions as title of paper
A good title is brief and adequately describes methods, and or purpose of study
Introduction:
Clearly state the research problem and highlight why it needs to be addressed
Describe population health management system and tools in simple words and briefly explain process evaluation
Methods:
The study uses both qualitative and quantitative methods.
Qualitative methods are properly described
however , quantitative methods need to be explained , such as , how quantitative data will be collected?
Which data collection instrument will be used ?
Describe data collection instrument
Ethical Concerns :
The study was granted approval by UCL Ethics committee , is not enough Describe how informed consent will be obtained ?
How confidentiality issues will be addressed?
Discussion :
Discussion should include a paragraph on significance of study
Conclusion:
Show how your introduction and conclusion are interlinked
Author Response
thank you to the reviewer for these helpful comments. We attach our point by point response here.

Reviewer 4 Report
1) Based on the title and the abstract, I thought I was going to review a protocol for data collection for COVID-19 disparities that has already occurred. But that is not the case here. Maybe the title needs to be reconsidered. Or would/should this protocol be part of the supplement of the actual results of the evaluation? I think this is an important topic worldwide, but this would be better presented with results, yes? Or maybe that is the idea to publish the protocol first and then publish the results. Speaking of, I don't see a timeframe for the conclusion of the study/protocol.
2) There is a lot of description in the article with few pictures. It is a hard read to get through with all of the description of the tool itself, and I'm not sure of it's use without collection and presentation of tool/clinical data. That is, unless the protocol is to be published alone first, and then a second publication will occur with the collected data/evaluations.
3) There are 2 "aims" of the study starting on line 70, but then there are 3 specific objectives listed in lines 141-149. Are the objectives part of the "aims?" Also, in line 153, it says "to achieve the study's capacity building objectives," but in the objectives listed in lines 141-149, I don't see a capacity-building objective. "Capacity" is mentioned in one of the aims, but that describes a general approach to evaluating PHM systems. It seems like the "capacity building" part of this protocol is something additional and "in the future" as there is a separate evaluation process for looking at such PMH systems. Perhaps this description/study/protocol should just focus on the evaluation of the Dashboard, users, and eventual data.
Author Response
thank you to the reviewer for these helpful comments. We attach our point by point response to them here.

Round 2
Reviewer 4 Report
I am not sure if the download of the manuscript created errors or if the additional material was not well proofread; this time, I had to concentrate more on improving the writing style, flow, and grammar before being able to understand what was added/written. I would need to re-read the article again once these issues were fixed.
Lines 67-69: unclear - please revise
Lines 99/100: change to: "in phases starting in December 8 2020. The initial target population for vaccination was..."
Line 108: ethnic inequities...in what? In vaccination rates? Please add/revise, if that is what is meant.
Line 150: add inequities?
Lines 151-154: unclear; please revise
Line 158: capital t (To)
Line 158: working closely with
Line 159: locally
Line 160: future evaluations...of what? pandemics? public health issues?
Line 185: no comma needed
Line 213: enabled
Line 215: define NCL
Line 230: developed by GW, CM, and FA
Lines 245-248: Break up sentence/revise
Line 266: depend on the...?
Line 267: change "fed back" to used; do not capitalize dashboard
Line 302: Have been used? Should this be "were used" to to maintain verb tense consistency in the sentence?
Line 321: Nothing in the discussion is mentioned about the protocol. It talks more so about the evaluation, and the word "evaluation" is used quite often. Maybe in this first sentence add something about how the protocol was used for the overall evaluation process. I would look
Line 322: reduce inequities...in what?
Line 323: change to "achieving this aim will provide"
Author Response
thank you very much for these comments. It appears the version the reviewer accessed has been corrupted because some of the problems listed do not feature in our version. We have attached our responses and with the updated tracked file file, we have now also included a PDF with all changes accepted for ease of reading.
